# Tools and methods for high-throughput single-cell imaging with the mother machine

**Ryan Thiermann[1†], Michael Sandler[1†], Gursharan Ahir[1†], John T Sauls[1†], Jeremy Schroeder[2†], Steven Brown[1], Guillaume Le Treut[3], Fangwei Si[4], Dongyang Li[5], Jue D Wang[6], Suckjoon Jun[1]\***

[1]Department of Physics, University of California, San Diego, La Jolla, United States; [2]Department of Biological Chemistry, University of Michigan Medical School, Ann Arbor, United States; [3]Chan Zuckerberg Biohub, San Francisco, United States; [4]Department of Physics, Carnegie Mellon University, Pittsburgh, United States; [5]Division of Biology and Biological Engineering, California Institute of Technology, Pasadena, United States; [6]Department of Bacteriology, University of Wisconsin–Madison, Madison, United States

*For correspondence:
s2jun@ucsd.edu

†These authors contributed equally to this work

Competing interest: The authors declare that no competing interests exist.

**Abstract** Despite much progress, image processing remains a significant bottleneck for high-throughput analysis of microscopy data. One popular platform for single-cell time-lapse imaging is the mother machine, which enables long-term tracking of microbial cells under precisely controlled growth conditions. While several mother machine image analysis pipelines have been developed in the past several years, adoption by a non-expert audience remains a challenge. To fill this gap, we implemented our own software, MM3, as a plugin for the multidimensional image viewer napari. napari-MM3 is a complete and modular image analysis pipeline for mother machine data, which takes advantage of the high-level interactivity of napari. Here, we give an overview of napari-MM3 and test it against several well-designed and widely used image analysis pipelines, including BACMMAN and DeLTA. Researchers often analyze mother machine data with custom scripts using varied image analysis methods, but a quantitative comparison of the output of different pipelines has been lacking. To this end, we show that key single-cell physiological parameter correlations and distributions are robust to the choice of analysis method. However, we also find that small changes in thresholding parameters can systematically alter parameters extracted from single-cell imaging experiments. Moreover, we explicitly show that in deep learning-based segmentation, 'what you put is what you get' (WYPIWYG) – that is, pixel-level variation in training data for cell segmentation can propagate to the model output and bias spatial and temporal measurements. Finally, while the primary purpose of this work is to introduce the image analysis software that we have developed over the last decade in our lab, we also provide information for those who want to implement mother machine-based high-throughput imaging and analysis methods in their research.

## eLife assessment

This article provides a review and test of image-analysis methods for bacteria growing in the 'mother-machine' microfluidic device, introducing also a new graphical user interface for the computational analysis of mother-machine movies based on the 'Napari' environment. The tool allows users to segment cells based on two previously published methods (classical image transformation and thresholding as well as UNet-based analysis), with **solid** evidence for their robust performance based

on comparison with other methods and use of datasets from other labs. While it was difficult to assess the user-friendliness of the new GUI, it appears to be **valuable** and promising for the field.

## Introduction

The mother machine (*Wang et al., 2010*) is a popular microfluidic platform for long-term, high-throughput imaging of single cells. It has been widely adopted as a standard for long-term imaging of bacteria such as *Escherichia coli* and *Bacillus subtilis* (*Sauls et al., 2019a*), as well as the eukaryote *Schizosaccharomyces pombe* (*Nakaoka and Wakamoto, 2017*; *Spivey et al., 2017*). In the mother machine, thousands of single cells are trapped in one-ended growth channels that open into a central trench (*Figure 1-1.1*). The cells at the end of the growth channels (mother cells) grow and divide over hundreds of generations, while their progeny are successively flushed out of the device (*Figure 1-1.2, 1.3*). Data gathered from the mother machine has brought critical insight into diverse domains such as aging (*Wang et al., 2010*), single-cell physiology (*Jun et al., 2018*), starvation adaptation (*Bakshi*

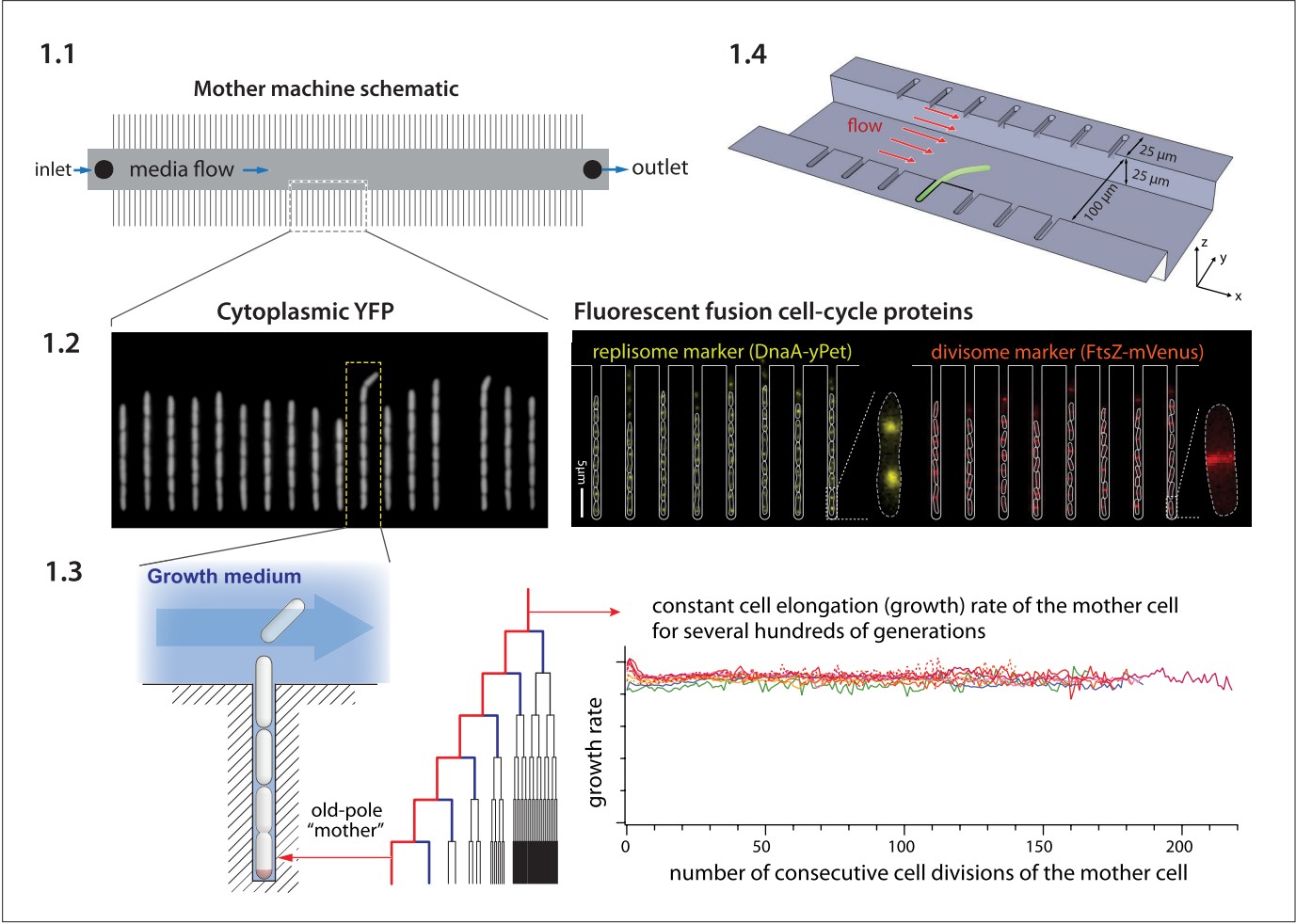

**Figure 1.** Mother machine workflow, schematic, and applications. (**1.1**) Mother machine schematic. Growth channels flank a central flow cell that supplies fresh media and whisks away daughter cells. In a typical experiment, numerous fields of view (FOVs) are imaged for several hours. (**1.2**) Fluorescence images of *E. coli* strains expressing cytoplasmic YFP (*Wang et al., 2010*) (left) and markers for the replisome protein DnaN and division protein FtsZ (right) (*Si et al., 2019*). (**1.3**) The mother machine setup allows long-term monitoring of the old-pole mother cell lineage (*Wang et al., 2010*) and has other versatile applications, including (**1.4**) the study of the mechanical properties of bacterial cells by applying controlled Stokes forces (*Amir et al., 2014*).

The online version of this article includes the following figure supplement(s) for figure 1:

**Figure supplement 1.** Inexpensive fabrication of cell loader with 3D printing.

*et al., 2021*), antibiotic persistence (*Kaplan et al., 2021*), cell differentiation (*Russell et al., 2017*), and the mechanics of cell wall growth (*Amir et al., 2014*; *Figure 1-1.4*).

Despite the progress in imaging techniques and microfluidics, image processing remains a major bottleneck in the analysis pipelines. The unique structure of the mother machine device enables precise control of growth conditions and long-term tracking of cells, to the degree that cannot be achieved by traditional tracking of cells in microcolonies (*Stewart et al., 2005*). However, automated image processing is essential to process the large amounts of data generated by these high-throughput experiments. In addition, the unique structure of the mother machine device requires a specialized workflow to select and track individual growth channels. As experimentalists often need to extract precise statistics over multiple generations or observe rare events, the analysis workflow must be modular to allow inspection and curation of intermediate results. To meet these needs, numerous mother machine-specific image analysis packages have been introduced in the last few years (*O'Connor et al., 2022*; *Sachs et al., 2016*; *Smith et al., 2019*; *Ollion et al., 2019*), in addition to general image analysis packages adaptable to the mother machine workflow (*Stylianidou et al., 2016*; *Panigrahi et al., 2021*; *Cutler et al., 2022*; *Spahn et al., 2022*; *Schwartz et al., 2019*). Much recent work has been catalyzed by advances in biomedical image analysis with deep convolutional neural networks (CNNs), particularly the U-Net architecture (*Ronneberger et al., 2015*). Many of these tools (*Ollion et al., 2019*; *Lugagne et al., 2020*) have been designed with ease-of-use and accessibility in mind. However, they can still present a steep learning curve for first-time users. In addition, as the outputs of these pipelines are often used by researchers to derive biological principles based on correlations, it is important to understand the limitations of and differences between different image analysis methods.

This article consists of three parts. First, for first-time users, we provide a brief walkthrough on implementing the mother machine in research (*Box 1*), including how to duplicate microfluidic devices at no cost using epoxy replicas and troubleshoot common image analysis problems. Next, we introduce MM3 (*Napari hub, 2023*), a fast and interactive image analysis pipeline for mother machine experiments that we have developed and used internally for over a decade. Our latest version is a Python plugin for the multidimensional image viewer napari (*napari contributors, 2023*). Finally, we compare the accessibility, performance, and robustness of various current image analysis platforms. In order to trust analysis results, researchers should understand the limitations of their chosen method. With this in mind, we show that 'what you put is what you get': both classical and deep learning-based segmentation methods are highly sensitive to user-determined threshold values. As exact cell boundaries may be difficult to distinguish by eye, these values are difficult to set definitively, and can systematically alter the output of the analysis. Fortunately, we find that key single-cell physiological parameter correlations and distributions are robust to the choice of analysis method. However, interpreting and comparing the results of different analyses require care.

## Results

### Mother machine image analysis with napari-MM3

Analysis of time-lapse imaging experiments requires dedicated software due to the sheer volume of data produced. For instance, an experiment tracking aging might require imaging 50 fields of view (FOVs; *Figure 1-1.1*) every 2 min for a week, producing a quarter of a million images comprising hundreds of gigabytes of data. While the experimental methods for mother machine experiments have become increasingly accessible, image analysis tools have lagged behind. Typically, labs using the mother machine have developed their own customized analysis pipelines. Many available tools require programming experience, familiarity with command line tools, and extensive knowledge of image analysis methods. They are also often fine-tuned for specific experimental setups and difficult for the average user to adapt. Finally, existing workflows frequently require users to move between multiple interfaces such as ImageJ, MATLAB, the command line, Python scripting, and Jupyter notebooks. Newer deep learning approaches are more versatile than traditional computer vision methods. Still, they bring new issues for novices: users may need to construct their own training data and train a model, requiring a new set of tools and technical expertise, and manual annotation of training data is susceptible to human error and bias.

# Box 1. Mother machine experimental workflow.

Despite the well-appreciated power of single-cell time-lapse imaging approaches, the potential user base remains much greater than the number of researchers directly benefiting from the technology. A primary reason for this discrepancy between demand and actual adoption is the perceived cost in time and resources of investment in the required core technology: microfluidics and high-throughput image analysis. Until a few years ago, setting up a typical microfluidic system for the first time took several years of training and trial-and-error, along with significant resources, for most individual labs.

Running a mother machine experiment requires the following steps: (1) fabricating a mold for the device, (2) assembling the device, (3) performing time-lapse microscopy, and (4) analyzing the images to extract time traces and statistics. To our knowledge, steps (1) and (4) have been the primary bottlenecks for most groups. Here, we give a brief overview of the experimental workflow. We refer interested readers to our previous review article on single-cell physiology (*Taheri-Araghi et al., 2015b*), along with other recent reviews (*Allard et al., 2022*; *Potvin-Trottier et al., 2018*) and published protocols (*Cabeen and Losick, 2018*), for a more extensive guide to single-cell imaging techniques.

**Device design and fabrication**. In the original mother machine design (*Wang et al., 2010*), narrow channels trap bacterial cells perpendicular to a larger main trench through which fresh medium flows (*Figure 1-1.4*). Several constraints apply to the design of the device. The height and width of the channels should match the dimensions of the organism under study. The channels must be large enough to facilitate the loading of the cells and allow for fast diffusion of nutrients to mother cells at the channel ends. If the channels are too deep, cells may move out of focus and potentially overlap in the *z*-direction, both of which impede accurate segmentation. Similarly, if channels are too wide, cells may not grow in a single file, complicating segmentation and tracking. Longer trenches will retain cells longer and allow more cells to be tracked per channel. The prohibitive cost of mold fabrication in clean room facilities has been a bottleneck to distributing microfluidic devices. We resolved this problem using an epoxy-based fabrication technique (*Kamande et al., 2015*), allowing us to easily and cheaply create replicative molds (*Box 1—figure 1*). Once the first microfluidic device is fabricated in the clean room, the epoxy duplication method allows us to reliably create and distribute high-fidelity device molds at a fraction of the cost of the initial fabrication. Undergraduate students in our lab routinely perform this procedure. To assist new users of the

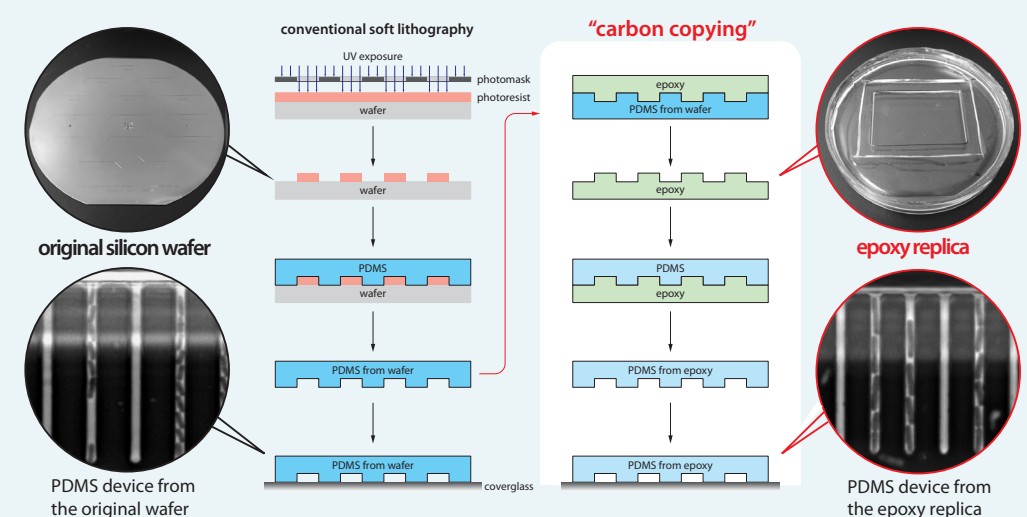

**Box 1—figure 1.** Duplication and distribution of mother machine devices with epoxy molds.

mother machine, we include a detailed procedure for the duplication method at *Thiermann, 2023*.

**Experiment setup**. The first step of making the mother machine device is to pour PDMS (polydimethylsiloxane) onto a master mold, cure it, and remove it from the mold. Holes are punched in the cut devices at the inlet and outlet of the central channel to connect tubing for fresh medium (inlet) and waste removal (outlet) before plasma treatment (*Figure 1-1.1*). Plasma treatment covalently bonds the PDMS device to a glass cover slide or dish to be mounted on the microscope. BSA (bovine serum albumin) passed through the device passivates the surface. In our setup, we load cells to the growth channels in the device via a custom centrifuge (*Thiermann, 2023*; *Figure 1—figure supplement 1*). Growth medium is passed through the device using a syringe pump. The medium flow should be fast enough to clear dead cells or biofilms in the device, but slow enough that the device does not delaminate. Mounting the device on an inverted microscope requires a custom stage insert for long-term imaging. The microscope temperature must be controlled tightly.

**Data analysis**. Most mother machine image analysis workflows share the following steps: preprocessing the acquired images, including identification and cropping of cell traps, cell segmentation, and cell tracking. Cell segmentation is the most difficult and crucial step, as adjacent cells must be separated from each other and from device features. After accurate segmentation, the one-dimensional structure of the mother machine – which constrains the cells to move only in one direction along the length of the trap without bypassing each other – makes cell tracking relatively simple.

These considerations guided us in the development of our in-house analysis tool. In building MM3, we sought to provide modularity and extensive interactivity while minimizing unnecessary user intervention. MM3 aims to be a complete and flexible solution for mother machine image analysis, taking raw images, and producing readily graphable cell data, while accommodating both machine learning-based and traditional computer vision techniques. It supports phase contrast and fluorescence images, and has been tested with different species (bacteria *E. coli* and *B. subtilis*, yeast *S. pombe*), mother machine designs, and optical configurations. The modular pipeline architecture allows flexible use of mid-stream outputs and straightforward troubleshooting (for instance, while *M. mycoides* is too small to segment with traditional microscopy methods (*Rideau et al., 2022*), we were able to obtain growth rate measurements by running the first half of the pipeline).

MM3 reflects the culmination of several iterations of our in-house mother machine analysis software developed over the past decade. Before MM3, we developed our image analysis pipeline in C++ (*Wang et al., 2010*) and MATLAB (*Taheri-Araghi et al., 2015a*). Eventually, Python became enormously popular, and we began MM3 as a set of Python scripts run from the command line (*Sauls et al., 2019b*). However, the command-line-based interface had several drawbacks. The interface was more difficult for users unfamiliar with the command line or programming. It also had limited interactivity. As a result, troubleshooting was difficult and required modifying the source code to display image output at intermediate steps or manually inspecting output files in ImageJ. This made the user repeatedly move back and forth between different windows and applications, slowing the analysis.

These drawbacks motivated us to convert MM3 into a plug-in for the Python-based interactive image viewer napari (*napari contributors, 2023*). napari provides an *N*-dimensional display ideal for visualizing multichannel time-lapse data. It offers built-in annotation tools and label layers to compare and annotate segmentation masks and tracking labels. It also provides a Python interpreter, allowing users to move easily between the viewer interface and the underlying data objects. For the best usability, we designed the napari-MM3 plug-in to allow the user to run the entire pipeline without leaving the napari interface.

Image analysis via napari-MM3 consists of four steps (*Figure 2* and *Figure 3*).

1. Crop raw images and compile them into stacks corresponding to individual growth channels.
2. Choose channel stacks to be (1) analyzed, (2) used as templates for background subtraction, or (3) ignored.
3. Segment cells.

4. Construct cell lineages. napari-MM3 treats individual cells in the lineages as objects that can be plotted directly or converted to another data format.

We elaborate on these steps as follows.

## Channel detection and curation

The first section of the napari-MM3 pipeline takes in raw micrographs and returns image stacks corresponding to one growth channel through time. napari-MM3 detects channels using a wavelet transform and then aligns them over time to correct for stage drift and vibration. The aligned growth channels are saved as unique image stacks with all time points for a given growth channel and color channel. As not all growth channels contain cells, napari-MM3 auto-detects channels as full or empty

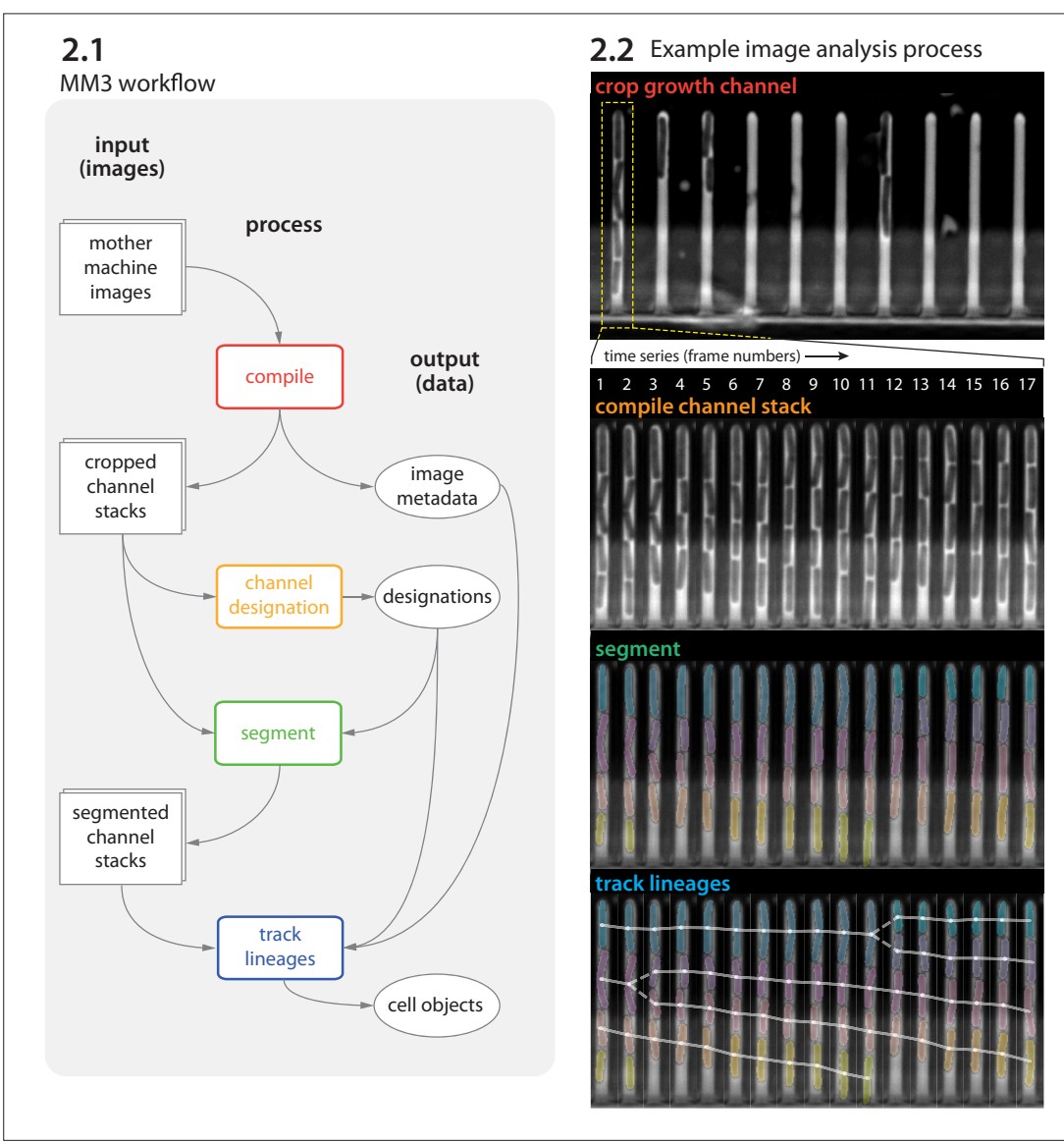

**Figure 2.** MM3 workflow and example images. (**2.1**) The MM3 image analysis pipeline takes raw mother machine images and produces cell objects. Processes (rounded rectangles) are modular; multiple methods are provided for each. (**2.2**) Example images from the processing of one growth channel in a single field of view (FOV). The growth channel is first identified, cropped, and compiled in time. All cells are segmented (colored regions). Lineages are tracked by linking segments in time to determine growth and division (solid and dashed lines, respectively), creating cell objects.

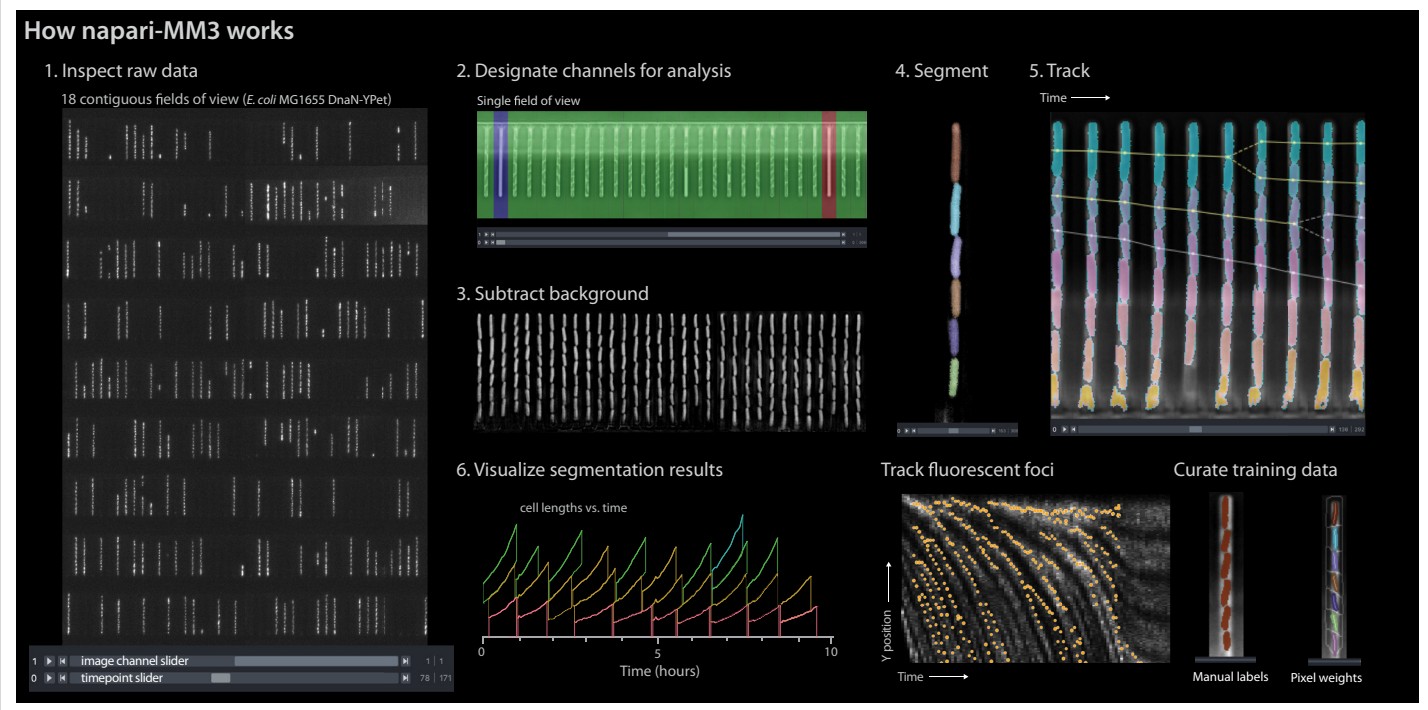

**Figure 3.** napari-MM3 interface. The napari viewer enables interactive analysis of mother machine data with real-time feedback and fast debugging. Raw data shown is from MG1655 background *E coli* expressing the fluorescence protein YPet fused to the replisome protein DnaN (*Si et al., 2019*).

based on the time correlation of the y-profile of the growth channel. The auto-detected growth channels and their classifications are then displayed in the napari viewer for the user to inspect and modify as needed.

## Cell segmentation

napari-MM3 offers two methods for cell segmentation, one using traditional computer vision techniques and the other using deep learning. The non-learning method utilizes Otsu's method to apply a binary threshold to separate cell objects from the background. It then labels the isolated cells and uses a random walker algorithm (*Grady, 2006*) to fill out the cell boundaries. This method is fast but optimized for specific mother machine designs and phase contrast imaging of bacteria. It also requires accurate background subtraction of phase contrast images (*Box 2*), to ensure that the presence of the channel border does not interfere with cell detection. The supervised learning method uses a CNN with the U-Net architecture (*Ronneberger et al., 2015*; *Lugagne et al., 2020*; *Falk et al., 2019*). The napari viewer can be used to construct training data, with the option to import existing Otsu or U-Net segmentation output as a template. The neural net can then be trained directly from napari, with the option to check the performance of the model in the napari viewer after successive rounds of training.

## Cell tracking and lineage reconstruction

Finally, napari-MM3 links segmented cells in time to define a lineage of cell objects, using a simple decision tree based on a priori knowledge of binary fission and the mother machine. Tracking produces a dictionary of cell objects containing relevant information derived from the cell segments, including the cell lengths and volumes over time, cell elongation rate, and generation time. Plotting and additional analysis can then be done with the user's tool of choice. Statistics can be directly extracted from the cell objects, or the cell objects can be converted into a.csv file, a pandas DataFrame, or a MATLAB structure. We provide a Jupyter notebook demonstrating this analysis at *Thiermann et al., 2024a* (copy archived at *Thiermann et al., 2024b*).

## Box 2. Segmentation via Otsu's method.

The Otsu segmentation method first aligns the growth channel of interest with an empty background channel by computing the orientation that maximizes the pixel-wise cross-correlation (*Box 2—figure 1*). The empty channel is then subtracted from the full channel, and the image is inverted. This background subtraction step is essential, as it removes the dark image of the PDMS device, which will otherwise interfere with segmenting the (dark) cells. Otsu's method (*Otsu, 1979*) is applied to find the binary threshold value that maximizes the inter-region variance. We then apply a Euclidean distance transform, wherein each pixel is labeled with its distance to the dark region. The image is thresholded again, and a morphological opening is applied to erode links between regions. Small objects and objects touching the image border are removed. Each region is labeled, and the labels are used to seed a random walker algorithm (*Grady, 2006*) on the original image.

Background subtraction and segmentation via Otsu's method and random walker algorithm

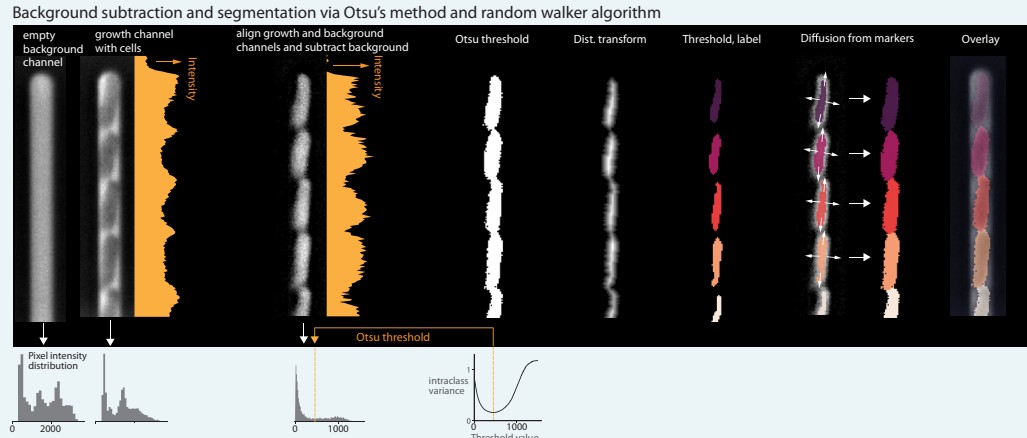

**Box 2—figure 1.** Background subtraction and segmentation via Otsu's method and random walker algorithm.

## Additional features and future extensions

napari-MM3 offers several additional modules supplemental to the main processing pipeline, including methods for fluorescence image analysis and U-Net training data construction and model training. Integrated fluorescence signal and fluorescence per cell area and volume for each time point can be extracted using the 'Colors' module. napari-MM3 also includes a module for the detection and tracking of fluorescent spots or 'foci'. For example, we have used it to track fluorescently labeled replisome machinery in bacteria in order to measure the timing and synchrony of DNA replication initiation (*Sauls et al., 2019a*; *Si et al., 2019*). Lastly, U-Net segmentation training data can be constructed by manual annotation of raw images in the napari viewer. napari-MM3 offers the option to construct training data with existing Otsu or U-Net segmentation data as a template. This allows the user to iteratively train a model, correct mistakes in its output, and use the modified output as input for the next round of training. We also provide a Jupyter notebook covering training data construction and model training at *Thiermann et al., 2024a*.

Going forward, we plan to add support for additional segmentation and tracking modalities (*Cutler et al., 2022*; *Ollion and Ollion, 2020*). We will also incorporate support for additional organisms such as the budding yeast *S. cerevisiae*. Finally, we plan to take advantage of napari's interactive display to add interactive data visualization and plotting.

**Table 1.** Performance metrics for napari-MM3.

Processing times were measured on an iMac with a 3.6-GHz 10-Core Intel Core i9 processor with 64 GB of RAM and an AMD Radeon Pro 5500 XT 8 GB GPU. Tensorflow was configured to use the AMD GPU according to *Apple Inc, 2023*. The GPU was used in U-Net training and segmentation steps. The dataset analyzed is from *Si et al., 2019* and consists of 26 GB of raw image data (12 hr, 262 time frames, 2 imaging planes, 34 fields of view [FOVs], and ~35 growth channels per FOV). Note that while the Otsu segmentation method is slightly faster than the U-Net, it also requires a background subtraction step, such that the total runtimes of the two methods are comparable.

| | Channel detection | Background subtraction | Segmentation (Otsu) | Segmentation (U-Net) | Tracking | Total (Otsu) | Total (U-Net) |
|---|---|---|---|---|---|---|---|
| Frame processing time | N/A | 2 ms | 4 ms | 5.3 ms | N/A | N/A | N/A |
| Channel stack processing time (262 time frames) | N/A | 0.54 s | 1.14 s | 1.4 s | 0.7 s | 3.1 s | 2.1 s |
| FOV processing time (35 channels) | 14.1 s | 17.5 s | 36.5 s | 46 s | 46.7 s | 2 min | 1.7 min |
| Exp. processing time (26 GB, 34 FOVs, ~20,000 cells) | 3.2 min | 9.9 min | 20.6 min | 26 min | 26.4 min | 60 min | 55 min |

## Performance test of napari-MM3

To evaluate the speed of napari-MM3, we timed the processing of a typical dataset (*Table 1*). Using consumer-grade hardware, a single-channel stack consisting of several hundred time frames can be processed in less than 5 s, and a typical experiment consisting of 25 GB of imaging data can be processed in under an hour. These metrics are on par with those reported by other recently published mother machine software (*Ollion et al., 2019*; *Lugagne et al., 2020*; *Banerjee et al., 2020*).

## Testing napari-MM3 on other published datasets

We tested napari-MM3 on several publicly available mother machine datasets: three from experiments on *E. coli* provided with the mother machine image analysis tools DeLTA, MoMA, and BACMMAN (*Ollion et al., 2019*; *Lugagne et al., 2020*; *Jug et al., 2014*) and one from *C. glutamicum* provided with the software molyso (*Sachs et al., 2016*). We were able to process all four datasets with minimal adjustments to the default parameter values (Methods). We quantified the performance of MM3 on each dataset by comparing the output of the MM3 segmentation to manually determined ground truth masks from a subset of each dataset (*Table 2*). To evaluate the segmentation quality, we computed

**Table 2.** Testing napari-MM3 on external datasets.

Quality of segmentation masks produced by running napari-MM3 on a subset of published datasets from other groups (*Sachs et al., 2016*; *Ollion et al., 2019*; *Lugagne et al., 2020*; *Jug et al., 2014*). As exact boundaries are difficult to determine by eye, we considered a cell to be correctly segmented if the Intersection over Union of the predicted mask and ground truth mask was greater than 0.6 (Methods). To evaluate the quality of the segmentation, we report the Jaccard index (*Laine et al., 2021*; *Taha and Hanbury, 2015*).

| Dataset | Correctly segmented cells | False positives | False negatives | Jaccard index |
|---|---|---|---|---|
| Ollion et al. (BACMMAN) (*Ollion et al., 2019*) | 228 | 4 | 1 | 0.98 |
| Lugagne et al. (DeLTA) (*O'Connor et al., 2022*; *Lugagne et al., 2020*) | 247 | 22 | 1 | 0.92 |
| Sachs et al. (molyso) (*Sachs et al., 2016*) | 247 | 4 | 0 | 0.98 |
| Jug et al. (MoMA) (*Jug et al., 2014*) | 80 | 0 | 0 | 1 |

**Table 3.** Overview of mother machine image analysis tools.

A comparison of several published imaging methods. '2D' or '1D' segmentation indicates whether the cells are labeled in an image and analyzed in two dimensions, or projected onto a vertical axis and analyzed in one dimension. Several tools support the use of deep learning (in place of or in addition to classical computer vision techniques).

| Software | Implementation | Segmentation | Deep learning support |
|---|---|---|---|
| BACMMAN *Ollion et al., 2019*/DistNet *Ollion and Ollion, 2020* | ImageJ plugin | 2D | ✓ |
| DeLTA *O'Connor et al., 2022*; *Lugagne et al., 2020* | Python package | 2D | ✓ |
| napari-MM3 *Sauls et al., 2019b*, this work | napari plug-in | 2D | ✓ |
| SAM *Banerjee et al., 2020* | MATLAB | 2D | |
| MMHelper *Smith et al., 2019* | ImageJ plugin | 2D | |
| molyso *Sachs et al., 2016* | Python package | 1D | |
| MoMA *Jug et al., 2014* | ImageJ plugin | 1D | |

the Jaccard index (JI) (*Laine et al., 2021*; *Taha and Hanbury, 2015*) at an intersection-over-union (IoU) threshold of 0.6 (Methods). The software performed well on the Ollion et al., Sachs et al., and Jug et al. datasets with JI of 0.98, 0.98, and 1, respectively. Segmentation was notably worse on the Lugagne et al. dataset, with JI of 0.92. However, we observed that most segmentation errors in the Lugagne et al. dataset arose from misclassification of cells near the channel opening, where determining cell boundaries is often more difficult.

## Comparison with other image analysis software

We also tested napari-MM3's usability and performance against other popular software. We began by surveying a range of existing mother machine image analysis tools (*Table 3*). Some early analysis pipelines used one-dimensional segmentation methods (*Sachs et al., 2016*; *Jug et al., 2014*), which perform adequately when cells are tightly confined in the growth channels. In recent years, many excellent general-purpose CNN-based cell segmentation tools have also been developed (*Stylianidou et al., 2016*; *Panigrahi et al., 2021*; *Cutler et al., 2022*; *Spahn et al., 2022*; *Stringer et al., 2021*), which may be extended to process mother machine data.

In this work, we only tested mother machine-specific pipelines. We constrained our analysis to DeLTA and BACMMAN, two excellent open-source mother machine-specific pipelines offering 2D segmentation and cell tracking, which are also well documented and actively maintained. BACMMAN (*Ollion et al., 2019*) performs 2D segmentation via traditional computer vision methods similar to those implemented in napari-MM3 and has recently added support for CNN-based segmentation as well (*Ollion et al., 2013*). DeLTA (*O'Connor et al., 2022*; *Lugagne et al., 2020*) uses the U-Net architecture for channel detection, cell segmentation, and cell tracking, with a mother machine-specific and general agar pad mode. We used BACMMAN, DeLTA, and napari-MM3 to analyze the same published dataset (*Si et al., 2019*; *Thiermann et al., 2024c*) consisting of *E. coli* MG1655 grown in minimal growth medium (MOPS modified buffer + 0.4% glycerol + 11 amino acids with ~60 min doubling time). Data processed in napari-MM3 was separately segmented with U-Net and traditional computer vision methods. We found that the pre-trained mother machine model provided with DeLTA did not generalize well to our data. However, after training a new model with representative data, we achieved accurate segmentation.

We compared the distributions and correlations of key physiological parameters generated by each analysis tool, motivated by our standard approach to single-cell physiology (*Jun et al., 2018*; *Taheri-Araghi et al., 2015a*; *Si et al., 2019*; *Le Treut et al., 2021*). First, we confirmed that all four analysis methods yield essentially identical correlations between cell length at birth ($S_B$) vs. (1) generation time ($\tau$), (2) elongation rate ($\lambda$), and (3) the length added between birth and division ($\Delta$) (*Figure 4-4.3*). Next, we compared the distributions of various physiological parameters. The CV (coefficient of variation) of a physiological parameter distribution is often taken to reflect the tightness of the underlying biological control. We have previously found (*Sauls et al., 2019a*; *Taheri-Araghi et al., 2015a*) that the CVs of a set of physiological parameters (birth length, division length, length added between divisions, growth rate, generation time, and septum position) are invariant across growth conditions in *E. coli*

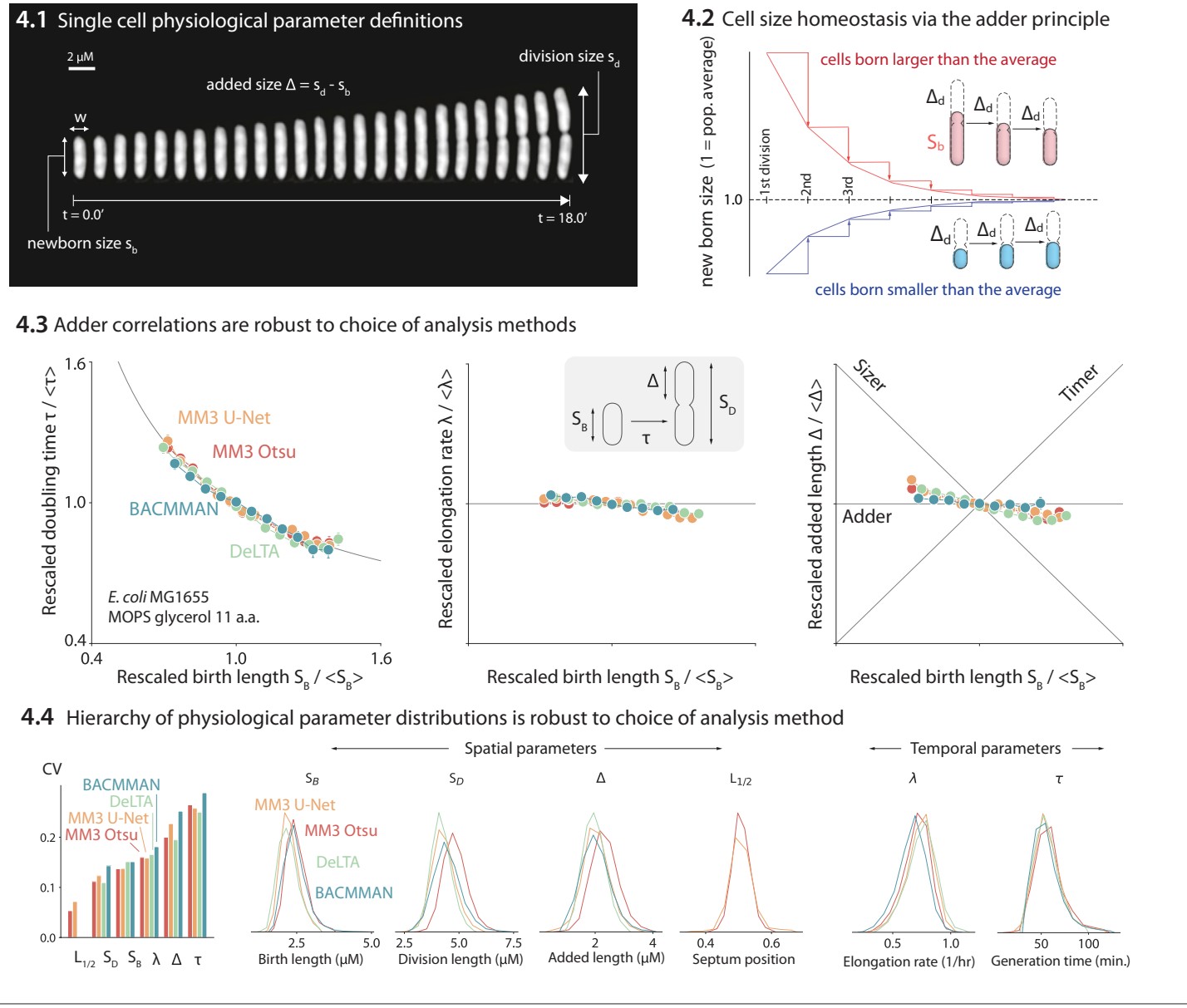

**Figure 4.** Comparison of various image analysis approaches. (**4.1**) A time series of a typical cell growing in a nutrient-rich medium. The birth size, division size, and added size are indicated. (**4.2**) The adder principle ensures cell size homeostasis via passive convergence of cell size to the population mean. (**4.3**) We analyzed multiple datasets from our lab using MM3, DeLTA, and BACMMAN, and obtained robust correlations between birth length, doubling time, elongation rate, and added length. Representative results from one dataset (**Si et al., 2019**) for MG1655 background *E. coli* grown in MOPS glycerol + 11 amino acids are shown, with 9000–13,000 cells analyzed depending on the method. Error bars indicate standard error of the mean (note the standard error is smaller than marker size in most cases). (**4.4**) Distributions of key physiological parameters are independent of the analysis methods. The data and code used to generate this figure are available at **Thiermann, 2024a**.

The online version of this article includes the following figure supplement(s) for figure 4:

**Figure supplement 1.** Old-pole aging phenotype is strain specific.

and *B. subtilis*, and that the hierarchy of CVs is preserved across the two evolutionarily divergent species (**Sauls et al., 2019a**; **Taheri-Araghi et al., 2015a**). Here, we confirmed that the distributions of these physiological parameters are independent of the analysis methods (**Figure 4-4.4**). In particular, the hierarchy of CVs is preserved by all three methods tested. Last, while in this dataset the old-pole 'mother' cells showed signs of aging (in particular, a reduced elongation rate), this aging phenotype is strain- and condition-dependent (**Figure 4—figure supplement 1**).

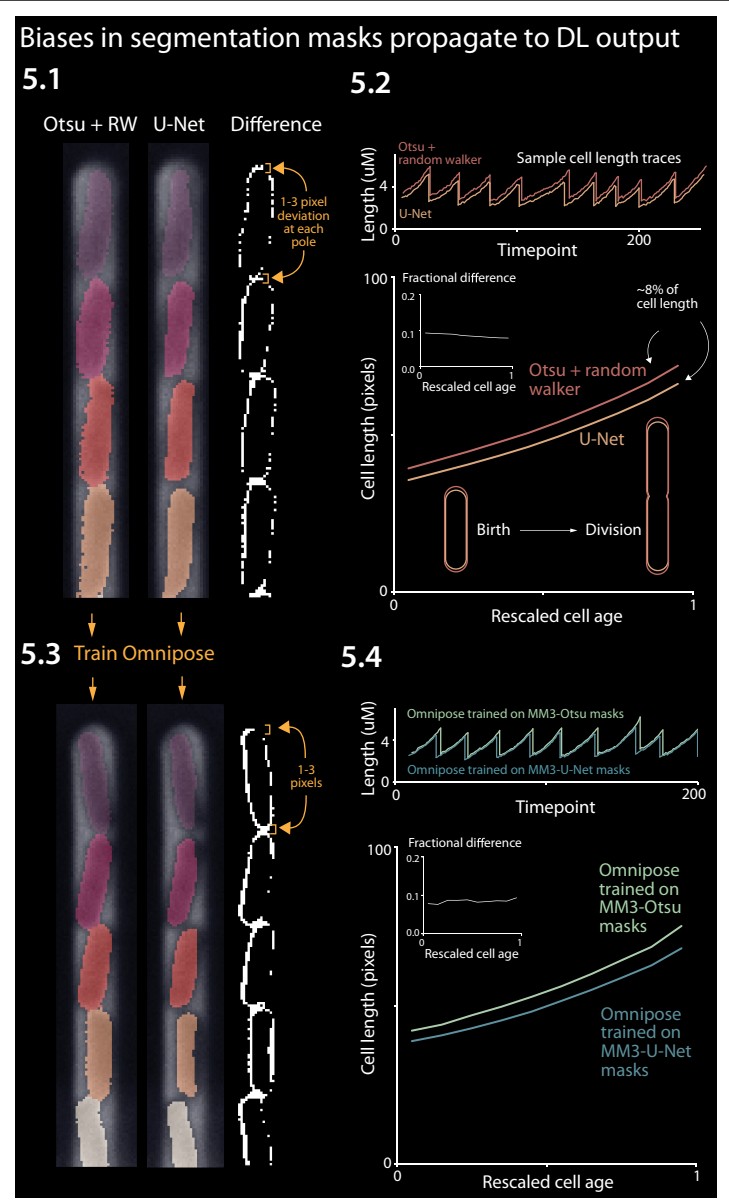

**Figure 5.** Effect of systematic deviation in segmentation output from different methods. (**5.1**) Otsu/random walker and U-Net segmentation masks. The classical method systematically yields masks that are 5–10% larger than the other methods. (**5.2**) We confirmed that this discrepancy occurs consistently across the cell cycle. (**5.3**) We trained the Omnipose model on masks generated by either napari-MM3-Otsu or napari-MM3-U-Net separately. (**5.4**) The systematic discrepancy in the training data masks propagated to the output of the trained models.

The online version of this article includes the following figure supplement(s) for figure 5:

**Figure supplement 1.** Evaluating segmentation output of napari-MM3 Otsu and U-Net methods.

## Systematic discrepancies in cell segmentation outputs

While we found that the correlations between physiological parameters were preserved across the different analysis methods (*Figure 4-4.3*), we also observed systematic discrepancies in the results obtained by different methods, including cell length at birth ($S_b$), length at division ($S_d$), and length added between birth and division ($\Delta$) (*Figure 4-4.4*). In particular, napari-MM3's classical segmentation method systematically generated larger cell masks than napari-MM3 U-Net, DeLTA, and BACMMAN (*Figure 4-4.4*). We focused on the discrepancies between the two MM3 outputs. Although the deviation between the two masks may not appear significant when individual masks are inspected by

eye (*Figure 5-5.1*, *Figure 5—figure supplement 1*), the classical method yields cells that are 5–10% larger at each time point than those returned by the U-Net method when averaging over an entire experiment with tens of thousands of cells tracked (*Figure 5-5.2*). Cell birth and division times are also systematically shifted in the classical method, as the expanded cell boundaries lead the algorithm to split cells one to two time frames later on average.

The root of this discrepancy is as follows. Exact cell boundaries are difficult to distinguish by eye, and the classical methods tested here require the user to set threshold values that can systematically alter the measured cell size. Indeed, both MM3 and BACMMAN's non-learning method (which also uses Otsu thresholding and a watershed/diffusion algorithm) – output different cell masks with their 'default' parameter settings. On the other hand, binary U-Net segmentation methods, such as those implemented in napari-MM3 and DeLTA, tend to output smaller cell sizes because the model must leave a gap between cells so that they are not stitched together (note this is not a fundamental limitation of U-Net, but a consequence of our implementation: see, for example *Cutler et al., 2022* or *Ollion and Ollion, 2020* for more complex approaches which avoid this issue).

## WYPIWYG (What You Put Is What You Get) in deep learning-based image analysis

Given that classical methods are clearly sensitive to this threshold tuning, we predicted that deep learning approaches would also be impacted (*Laine et al., 2021*; *Geiger et al., 2021*). We chose the recent cutting-edge segmentation model Omnipose and separately trained it on masks derived from the Otsu segmentation output and masks from the napari-MM3 U-Net segmentation output. We chose Omnipose as it assigns different labels to different cells, and can thus segment cells with contiguous boundaries, in contrast to MM3 or DeLTA's U-Net implementations. Indeed, we found that the systematic discrepancy in the training masks propagated to the output of the trained models: the Omnipose model trained on larger Otsu masks generated larger masks upon evaluation with the same data, while the Omnipose model trained on smaller U-Net masks output smaller masks (*Figure 5-5.3*). In computer science, the phrase 'Garbage in, garbage out' denotes the concept that undesirable attributes in the input to a program will propagate to the output (*Mellin, 2024*; *Babbage, 1864*). Here, we propose a related notion WYPIWYG, or 'what you put is what you get'. That is, at least for our setup, systematic differences in training data masks lead the model to learn different threshold intensity values and thus to systematically output larger or smaller masks. We emphasize this result does not reflect a flaw in Omnipose – whose performance we found impressive – but rather a well-studied feature of machine learning methods in general (*Geiger et al., 2021*).

## Discussion

In this study, we introduced a modular and interactive image analysis pipeline for mother machine experiments and compared its effectiveness to other existing tools. Unlike its predecessors, napari-MM3 is equipped with an intuitive and modular interface, making it highly accessible to new users. Our main goal is to lower the barrier to entry in image analysis, which has been a primary obstacle in adopting the mother machine, and ultimately increase its user base.

Finally, we discuss common challenges faced by users new to high-throughput image analysis and give our prescriptions for overcoming them.

### Validating results

We showed that distributions and correlations in key cell cycle parameters are invariant to the choice of analysis pipeline, provided that care is taken in parameter adjustment and postprocessing. However, this parallel processing of data is not feasible for every experiment. Instead, we suggest users can validate their results in the following ways:

1. A qualitative 'eye test' is an important first step: one should always visually inspect one's data. Often, this may be sufficient to establish whether the analysis is operating as expected.
2. When a more quantitative and systematic approach is needed, the user can compare the output of their analysis to a subset of manually annotated 'ground truth' images. Quantitative measures such as the JI, $F_1$ score or dice coefficient may be used (*Laine et al., 2021*; *Taha and Hanbury, 2015*). These metrics are particularly useful for comparing the results of different parameter

choices in a given method, allowing the user to determine the combination that yields the most accurate segmentation or tracking results.

3. Verify that the averages calculated from single-cell measurements match the results of population-level control experiments.
4. When possible, filter for subsets of the data that are likely to reflect accurate segmentation and continuous tracking, such as cell lineages that are continuously tracked for the duration of the experiment.

## Choosing an image analysis tool

For many years, published and well-documented pipelines for mother machine image analysis were scarce, and existing software required extensive parameter reconfiguration, knowledge of image-processing techniques, and programming experience to use effectively. In recent years, advances in deep learning have contributed to a rapidly growing set of image analysis tools that perform cell segmentation and tracking.

Inspired by previous reviews (*Laine et al., 2021*; *Jeckel and Drescher, 2021*), we make the following suggestions for new users selecting a tool:

1. Tools that are actively maintained, with an easy way to contact the developer, will be more likely to work well and will be easier to troubleshoot than others.
2. Detailed documentation and tutorials are valuable and will allow the user to troubleshoot the software without direct guidance from the developers.
3. Depending on the user's level of comfort with coding, it may be beneficial to choose a tool that is implemented through a graphical user interface and does not require additional programming. Moreover, even for programmers, we found within our lab that introducing interactivity when necessary dramatically expedited the data analysis process.
4. Full stack (vertically integrated) tools that cover the entire analysis pipeline may save time and work, relative to those which only perform a portion of the needed analysis.
5. It is worthwhile to engage with the online community around the tool. We have found the image.sc forum (*Image.Sc, 2023*) valuable in the past for help with napari.
6. Consider whether the tool is open source or requires a license. We encourage tool developers to avoid proprietary software such as MATLAB, which may not be accessible to all users. The open-source Java-based image-processing program ImageJ (*Bourne, 2010*) has been a dominant tool in biological image analysis for many years. The recent growth of image analysis and machine learning tools in Python makes napari (*napari contributors, 2023*) an attractive alternative to ImageJ.

## Traditional computer vision vs. deep learning methods

A key choice many users will face is whether to use deep learning-based or traditional methods for image analysis. The field has increasingly shifted toward deep learning methods, and this shift will likely accelerate. While traditional computer vision methods remain useful, deep learning-based methods have a clear advantage in their ability to generalize quickly to new datasets.

In our lab, we have found that traditional computer vision techniques perform excellently on cell segmentation and tracking in the mother machine, subject to constraints on the experimental setup. However, such methods often require extensive reconfiguration or fail entirely when applied to data obtained under new biological conditions (different organisms, different cell morphology) and imaging conditions (varied illumination, microscope setup). Our own non-learning segmentation method performs well, provided that cells are tightly confined in the mother machine channels and do not move substantially. Prior to the adoption of deep learning methods, this requirement necessitated the design of different devices for cells grown in different growth conditions, as the cell width in some *E. coli* strain backgrounds varies with the population growth rate.

In contrast, the key strength of deep learning approaches is their ability to generalize to new conditions – whether to different illumination conditions, different types of input images (phase contrast, brightfield, fluorescence) or different organisms and cell types entirely. The main barrier to adoption of learning-based methods remains the construction of training data, which can be tedious and time consuming. A training data set of 50–100 images comprising several hundred cells can be constructed in a few hours and will achieve passable segmentation on representative data. However, larger training sets on the order of thousands of images are preferable and will yield improved model accuracy and

generalizability. The time needed for annotation can be reduced by seeding the data with masks generated by classical methods – or iteratively seeding with U-Net output – and then refining the masks further by hand. Model performance and generalizability can often be significantly improved by augmenting training data via manipulations such as rotating or shearing, distorting the intensity profile, and adding noise. Nonetheless, we have found that even with extensive data augmentation, applying the U-Net segmentation to new experimental configurations or imaging conditions often requires retraining the model on an expanded dataset with more representative data. Ultimately, deep learning methods are only as good as the data they are trained on and are most likely to fail when training data is insufficient, mislabeled, or not representative. Going forward, sharing of training sets and models (*Assets, 2023*) between different groups can facilitate progress and aid reproducibility.

In addition to deep learning-based segmentation, learning-based cell tracking in the mother machine has been implemented recently by multiple groups (*O'Connor et al., 2022*; *Ollion and Ollion, 2020*). For cells growing unconstrained on 2D surfaces such as agar pads, U-Net tracking dramatically outperforms traditional methods (*O'Connor et al., 2022*). On the other hand, for steady-state growth in the mother machine where cells are confined and constrained to move in one dimension, we have not found a significant difference between the performance of deep learning-based tracking and the non-learning tracking method implemented in MM3. In both cases, errors in tracking nearly always arise from errors in segmentation. Nonetheless, deep learning-based tracking may offer an advantage in cases where cells may move substantially along the length of the channel or undergo dramatic morphological changes such as filamentation.

Ultimately, for groups with existing analysis pipelines fine-tuned for specific organisms under specific imaging conditions to perform simple tasks such as segmentation and 1D tracking, there may be little incentive to switch to deep learning methods. However, for users looking to develop a new pipeline or analyze more complex data, the power and generality of deep learning tools will make them the method of choice.

## Should users worry about the systematic discrepancy in segmentation results between different methods?

Given the 5–10% variance in the segmented bacterial cell size is comparable to the CVs of several physiological parameters (*Figure 4*), should researchers be concerned about the robustness of their results? The answer depends on the purpose of the image analysis.

If the research critically relies on the absolute cell size, such as cell-size control (*Taheri-Araghi et al., 2015a*; *Si et al., 2019*), the researcher must be aware of inherent limitations to the accuracy of spatial measurements from cell segmentation. These arise in part from the difficulty of consistently distinguishing cell boundaries by eye. Once a threshold is chosen, the choice will affect all analyzed cells systematically. This limitation applies to both deep learning (through the construction of training data) and traditional computer vision methods (through the manual input of a threshold value). For cell segmentation, the uncertainties are typically comparable to the pixel size of the images, rather than optical resolutions. For example, the pixel size in the images in *Figure 5* is 0.065 µm (for the camera pixel size 6.5 µm and ×100 magnification), which is non-negligible for many commonly cultured bacterial cells with submicron cell widths – for example, *Enterobacterales*, *Pseudomonas*, *B. subtilis*, and *Caulobacter crescentus*. For most commercially available cameras and objective lenses used in quantitative bacterial cell biology, 10% should be taken as a conservative lower bound for uncertainty when comparing absolute spatial measurements of bacterial cell size.

Indeed, researchers should be particularly careful when comparing absolute measurements of cell size, for example, at division or initiation of chromosome replication obtained by different groups using different image analysis methods. While absolute temporal measurements are more robust than spatial measurements (*Figure 4-4.4*), the differences in spatial measurements can propagate to the measured timing of, for example, cell division. For instance, we observed that the classical method stitched cells together for slightly longer than the U-Net method did (*Figure 5-5.2*), but as this shift applied equally to birth and division, it did not affect the average cell generation time (*Figure 4-4.4*).

Fortunately, the examples mentioned above are extreme cases. For instance, the pixel-size uncertainties will reflect a smaller proportion of the cell size when imaging larger cells such as yeast or mammalian cells. Even in our research on single-cell bacterial physiology (*Sauls et al., 2019a*; *Taheri-Araghi et al., 2015a*; *Si et al., 2019*), we find that correlations and relative changes are more likely to

be robust than absolute spatial measurements to the choice of analysis method (*Figure 4*). Furthermore, different applications of deep learning-based image analysis, such as high-throughput phenotypic classification (*Shiaelis et al., 2023*) will be much more robust to the pixel-size uncertainties in image segmentation results.

## Generating robust and unbiased segmentation results

We have shown that both traditional computer vision and deep learning methods are susceptible to biases introduced by imprecise thresholding and human error. How, then, can more precise cell boundaries be determined? For non-learning methods, thresholds could be calibrated against data from alternate imaging methods such as fluorescence or brightfield. For learning methods, one promising technique is the generation of synthetic training data (*Hardo et al., 2022*). This method also has the advantage that new training datasets can be instantaneously for different imaging conditions or cell types once the appropriate parameters have been determined. For deep learning methods, metrics which lead the model to recognize cell interiors or centers (*Cutler et al., 2022*; *Ollion and Ollion, 2020*; *Naylor et al., 2019*) may yield more robust results than binary pixel-level classification. Once cell centers are known, boundaries can be determined relatively easily via classical watershed or random walker diffusion algorithms.

## Conclusion and recommendations

Here, we presented a guide to first-time users of the mother machine, introduced our updated image analysis software, and validated it against existing tools. napari-MM3 provides a simple and modular user-friendly interface, which we believe makes it uniquely accessible and valuable to novice users. By lowering the barrier to entry in image analysis – the key bottleneck in mother machine adoption – we aim to increase the user base of this powerful tool dramatically.

After testing two other well-constructed mother machine image analysis pipelines, we concluded that all four methods (BACMMAN, DeLTA, MM3 Otsu, and MM3 U-Net) yielded consistent and reproducible results, up to previously discussed limitations of segmentation algorithms. Thus, for users already comfortable with a given pipeline, there is no strong incentive to switch to a new one. However, the different pipelines do have markedly different user interfaces. DeLTA is set up to provide a simple 'one-shot' analysis, in which image preprocessing, channel detection, segmentation, and tracking are performed in sequence with minimal user input. This arrangement simplifies the analysis process, especially for first-time users. In particular, it can be helpful for users who want to quickly verify that the software will serve their purpose, before investing more time in setting up and running the analysis. On the other hand, the intermediate steps in the pipeline are less accessible, which may make debugging and troubleshooting more involved. BACMMAN, like napari-MM3, is more modular than DeLTA. This modularity can aid troubleshooting and improves versatility, but configuration can be time consuming. With napari-MM3, we attempted to strike a balance between these two well-designed and well-performing tools, while taking advantage of the fast-growing next-generation image analysis platform napari. napari-MM3 attempts to infer or pre-set as many parameters as possible, while the napari interface makes midstream output easily accessible. We have been using MM3, and more recently napari-MM3, for over a decade since our introduction of the mother machine in 2010, and we will continue to actively maintain and improve it in the coming years.

The mother machine setup has become increasingly accessible to researchers in recent years, through the distribution of molds and the publication of in-depth protocols and open-source image analysis software. At the same time, new variations of the device have found diverse applications, including bacterial starvation (*Bakshi et al., 2021*) and genetic screening (*Lawson et al., 2017*; *Luro et al., 2020*). Clearly, the combination of microfluidics with high-resolution time-lapse imaging remains powerful among single-cell techniques. We hope that this article will prove useful to mother machine veterans and first-time users alike.

## Methods
### Resources

- napari-MM3 Github repository (*Thiermann et al., 2024a*, copy archived at *Thiermann et al., 2024b*).

○ Contains installation instructions and video tutorial.
- Jupyter notebook demonstrating analysis of MM3 output data (*Thiermann et al., 2024a*).
  ○ A notebook providing functions for postprocessing and plotting of the napari-MM3 output.
- Protocols for device fabrication and loading (*Thiermann, 2023*).
- Raw data analyzed in this manuscript (*Thiermann et al., 2024c*).
- Processed data analyzed in this manuscript (*Thiermann, 2024a*, copy archived at *Thiermann, 2024b*).

## Getting started with napari-MM3

napari-MM3 is implemented entirely in Python and can be accessed on Github (*Thiermann et al., 2024a*), along with documentation covering installation and usage. It will run on a standard Mac, PC, or Linux machine. We recommend using the Anaconda Python distribution to simplify installation.

## Imaging conditions

The data analyzed in *Figures 4 and 5* (originally published in *Si et al., 2019*) was obtained on an inverted microscope (Nikon Ti-E) with Perfect Focus 3 (PFS3), ×100 oil immersion objective (PH3, numerical aperture = 1.45), and Obis laser 488 LX (Coherent Inc, CA) as a fluorescence light source, and an Andor NEO sCMOS (Andor Technology) camera. The laser power was 18 mW. The exposure time was 200 ms for phase contrast imaging and 50 ms for fluorescence.

## Image analysis for software comparison

For the software comparison in *Figure 4*, we analyzed a dataset from *Si et al., 2019* consisting of *E. coli* MG1655 expressing a fluorescent protein YPet fused to the replisome protein DnaN. The cells were grown in MOPS minimal medium + glycerol and 11 amino acids. The dataset was analyzed end-to-end starting from the raw.nd2 file with BACMMAN, DeLTA, and MM3. For analysis with DeLTA, we used the provided channel detection and tracking models but trained a new model on our own data for segmentation. For segmentation with BACMMAN, we used the standard non-learning phase contrast segmentation method 'MicrochannelPhase2D'. Postprocessing of the output of each pipeline was done in Python. For each pipeline, we filtered for cells whose mothers and daughters were also tracked.

The code and data to reproduce the plots in *Figure 4* are available at *Thiermann et al., 2024a* and *Thiermann, 2024a*, respectively. The raw image data is available at *Thiermann et al., 2024c*.

**Table 4.** MM3 parameter values for processed external datasets.
Parameters which were changed from the default values are shaded in yellow. Ollion et al., Jug et al., and Sachs et al. datasets were segmented with the non-learning method, while the Lugagne et al. dataset was segmented using the U-Net method.

| | | Default value | Ollion et al. | Lugagne et al. | Jug et al. | Sachs et al. |
|---|---|---|---|---|---|---|
| Compile | Channel width (px) | 10 | 20 | 10 | 10 | 10 |
| | Channel separation (px) | 45 | 90 | 45 | 45 | 45 |
| Subtract | Align pad (px) | 10 | 10 | 10 | 10 | 10 |
| | 1st opening (px) | 2 | 3 | N/A | 3 | 3 |
| | Distance threshold (px) | 2 | 3 | N/A | 3 | 3 |
| | 2nd opening (px) | 1 | 2 | N/A | 1 | 2 |
| | Otsu threshold scale | 1 | 1.2 | N/A | 1.0 | 1.0 |
| Segment | Min object size (px$^2$) | 25 | 25 | 25 | 25 | 25 |
| | Growth length ratio (min, max) | (0.8, 1.3) | (0.9, 1.5) | (0.8, 1.3) | (0.8, 1.3) | (0.8, 1.3) |
| | Growth area ratio (min, max) | (0.8, 1.3) | (0.9, 1.5) | (0.8, 1.3) | (0.8, 1.3) | (0.8, 1.3) |
| | Lost cell time (frames) | 3 | 3 | 3 | 3 | 3 |
| Track | New cell y cutoff (px) | 150 | 300 | 150 | 150 | 150 |

For the comparison of Otsu and U-Net outputs from Omnipose in *Figure 5*, we trained Omnipose with a learning rate of .01 without a pre-trained model. We used the same set of 1000 randomly selected images for both Otsu and U-Net, the only difference coming from the labeled masks themselves. Both models were trained until the loss dipped below 0.9 (390 epochs for U-Net, 210 epochs for Otsu). In some cases, the model 'hallucinated' cells along the channel features. We excluded these images from the final analysis.

## Analysis of external datasets

The external datasets were preprocessed as follows: Ollion et al., Jug et al., and Sachs et al. datasets were rotated 1–2 degrees to align the channels vertically. Ollion et al., Sachs et al., and Lugagne et al. datasets were cropped to remove imaging artifacts from the main trench.

The parameter values used for analysis of each dataset are shown in *Table 4*. In general, the optimal parameter values for the compilation and subtraction steps depend on the size of device features as well as the optical resolution and camera pixel size, while the optimal segmentation parameters depend on cell size as well as pixel size and optical resolution. Finally, the tracking parameters are either sensitive to the imaging frequency and the single-cell elongation rate (growth ratios and lost cell time), or the spatial position of the cells in the frame (y cutoff). The output cell size is sensitive to the 'Otsu threshold scale' parameter, so care should be taken when adjusting this value. In addition, the growth length and growth area ratio parameters may filter out fast- or slow-growing cells if they are set too close to 1. The remaining parameters will not impact the output statistics.

Each dataset was processed in its entirety with napari-MM3. To evaluate the segmentation quality, we selected one to two representative traps (comprising 50–100 time steps) and constructed ground truth masks for these images. On this subset, we computed the JI (*Taha and Hanbury, 2015*) as the ratio of true positives (correctly identified cells) to the sum of true positives, false positives (identified cells which were not present in the ground truth data), and false negatives (ground truth cells which were not identified by the segmentation). The segmentation and ground truth masks were determined to be matching if their IoU value was at least 0.6. Note that two masks become indistinguishable to the human eye at IoU 0.8 and higher (*Cutler et al., 2022*; *Laine et al., 2021*).

The output JSON file and kymographs showing reconstructed cell lineages from each sample datasets are available at *Thiermann, 2024a*, along with JSON files containing the parameter values used for each step of the analyses.

## U-Net model training

Training data was augmented as described below to aid the generalizability of the model. We trained the U-Net model using a binary cross-entropy loss function, with pixel-wise weighting to force the model to learn border pixels (*Ronneberger et al., 2015*; *Lugagne et al., 2020*). The model was trained using the Adam optimizer with a learning rate of $10^{-4}$, a dropout rate of 50%, a batch size of eight samples, a patience (early stopping value) of 50 epochs and a train-test split of 90–10.

## Overview of the MM3 pipeline

### Channel compilation and designation

The first section of the MM3 pipeline takes in raw micrographs and returns image stacks corresponding to one growth channel over time. Further pipeline operations are then applied to these stacks.

A standard mother machine experiment consists of thousands of images across multiple FOVs and many time points. Images are first collated based on the available metadata. MM3 expects TIFF files and looks for metadata in the TIFF header and from the file name.

All images from a particular FOV are analyzed for the location of channels using the phase contrast plane. Channel detection is performed using a wavelet transform, in which a mask is made which is applied across all time points. Channels are cropped through time using the masks and saved as unique image stacks that include all time points for a given channel and imaging plane. MM3 saves channel stacks in TIFF format.

MM3 attempts to compile all channels. However, not all channels contain cells, and some channels may have undesirable artifacts from the device preparation. It is, therefore, desirable to only process certain channels for analysis. Consequently, MM3 auto-detects empty and full channels based on the time correlation of the y-profile of the channel (empty channels are highly correlated in time,

while channels containing cells are not). The auto-detected channels and their classifications are then displayed in the napari viewer for the user to inspect and modify as needed. The user may also manually select empty channels free of artifacts to be used as templates for phase or fluorescence background subtraction.

## Background subtraction

MM3's Otsu segmentation method requires background subtraction of phase contrast images. The subtraction ensures that the presence of the channel border does not interfere with detection of cells. To this end, we overlay the previously identified empty channels on the full channels to be subtracted. The two channels are aligned such that the cross-correlation of overlaid pixels is maximized. After the inversion of the image, this leaves the cells as the only bright objects on a dark background. Good alignment of the device features in the empty and full channel is essential here. Imperfect alignment will leave artifacts in the subtracted image, which interfere with later steps, and is a common failure point for this method. Note that the subtraction step necessitates the presence of some empty channels in each experiment. The U-Net segmentation does not require background subtraction.

## Cell segmentation

Cell segmentation is the first of the two major tasks in the image analysis pipeline. Segmentation receives channel stacks and produces 8-bit segmented image stacks. Typically, segmentation is done using the phase contrast time-collated stack.

MM3 has two methods for segmentation: a 'standard' method and a supervised learning method. The standard method uses traditional image analysis techniques, specifically background subtraction, Otsu thresholding, morphological operations, and watershed algorithms. As the standard method may require fine-tuning of parameters, the napari plugin allows the user to quickly preview the effect of tuning morphological parameters and threshold value on the segmentation output, without having to process the entire dataset. The Otsu segmentation method first aligns the channel of interest with an empty background channel by computing the orientation, which maximizes the pixel-wise cross-correlation. The empty channel is then subtracted from the full channel, and the image is inverted. Otsu's method is then applied to find the binary threshold value which maximizes the inter-region variance (or equivalently, minimizes the intra-region variance). We then apply a Euclidean distance transform, in which each pixel is labeled with its distance to the dark region. The image is thresholded again, and a morphological opening is applied to erode links between regions. Small objects and objects touching the image border are removed. Each region is labeled, and the labels are used to seed a random walker algorithm (*Grady, 2006*) on the original image. As implemented in MM3, this 'standard' method has three adjustable parameters: the first opening pixel size, second opening pixel size, distance threshold (i.e., threshold which is applied to the distance transformed image, in pixels), and a dimensionless parameter to rescale the Otsu-determined threshold, if needed.

The supervised learning method uses a standard U-Net architecture with five levels (*Ronneberger et al., 2015*). The model outputs a cell class probability between 0 and 1 for each pixel, which is thresholded at 0.5 to obtain a binary segmentation. The napari viewer can be used to construct training data, with the option to import existing Otsu or U-Net segmentation output as a template. The neural net can then be trained using a separate widget, with the option to check the performance of the model in the napari viewer after successive rounds of training. We found that applying a weighted loss depending on pixel location – as suggested in the original U-Net paper (*Ronneberger et al., 2015*) and implemented for instance in DeLTA (*Lugagne et al., 2020*) – sped up model training and improved segmentation and tracking. Since the accurate separation of adjacent cells is vital for cell tracking, the cost of misidentifying pixels between bordering cells is high. We initially implemented a simple binary weight map where pixels between cells were weighted highly and all other pixels relatively lower. We later added a more complex mapping, drawing directly from the one implemented in DeLTA (*O'Connor et al., 2022*), where weights are maximized on the skeletons (*Lee et al., 1994*) of the cells and borders. Intuitively, this weighting tells the model that pixels in the center of the cell, in regions far from cells, and on the borders between cells are most important to predict accurately.

Illumination conditions can vary across laboratories, microbial species, and with device design. To aid the generalizability of the U-Net model, on specific conditions, we augmented the training data with various morphological techniques, including changing magnification, zooming, and rotating, and

Gaussian noise and blur. We also adapted several non-standard operations from DeLTA, one which performs elastic deformation and two others that distort image contrast to simulate changes in illumination within the FOV and between experiments.

### Cell tracking

Tracking segmented cells is the second major task in the pipeline. Tracking involves linking cell segments in time to define a lineage of cell objects. The default tracking method is a simple decision tree based on a priori knowledge of binary fission and the mother machine. For example, cells normally grow by a small amount between time intervals, divide into two similarly sized daughter cells, and cannot pass each other in the channel. The tracking method accounts for the absolute positions and relative ordering of cells in each channel over time. Specifically, at each time point we iterate over all detected regions (potential cells). Based on their relative y positions in the channel and sizes, each is linked to a set of potential descendants/ancestors. When two cells are best matched to the same region, the event is classified as a division, subject to constraints on the size of the regions. This tracking implementation is like that employed by BACMMAN (*Ollion et al., 2019*) although it does not explicitly consider relative ordering of cells in the channel. It contrasts with more complex optimization-based methods used by other mother machine software (*Sachs et al., 2016*; *Jug et al., 2014*).

The lineage tree obtained by tracking is displayed in the napari viewer in the form of a kymograph, in which the *x*-axis represents time, and cell linkages and divisions are indicated by forking lines.

### Data output and analysis

Tracking produces a dictionary of cell objects which contains relevant information derived from the cell segments. This includes, but is not limited to, birth and division size, growth rate, and generation time. Each object is identified by a key that represents the FOV and channel of the cell, the time point of its birth, and its position in the channel. Since each cell object has the requisite information to find its corresponding position in the channel stacks, the objects can be modified and extended by additional analysis. For example, the corresponding location of a cell in a fluorescent image stack can be retrieved, focus detection performed, and that information can be added to the cell object. This minimizes the burden of rerunning previous sections of the pipeline for new subanalyses.

Plotting can be done from this cell object dictionary directly, or it can first be converted to a.csv, a pandas DataFrame, or a MATLAB structure. We provide a Jupyter notebook (*Thiermann et al., 2024a*) to illustrate how the data can be extracted and plotted.

### Fluorescence analysis

Integrated fluorescence signal and fluorescence per cell area and volume for each time point can be extracted using the Colors module.

### Focus tracking

The focus tracking module enables the identification and tracking of fluorescent spots or 'foci'. This module has been used in our lab for tracking fluorescently labeled replisome machinery in bacteria to measure the timing and synchrony of DNA replication initiation. However, it may be applied to any use case requiring localization and tracking of intracellular spots. The module uses a Laplacian convolution to identify fluorescent spots. Foci are linked to the cell objects in which they appear.

### U-Net training data annotation

Training data can be constructed by manual annotation of raw images in the napari viewer. MM3 offers the option to construct training data with existing (Otsu or U-Net) segmentation data as a template. This allows the user to iteratively train a model, correct mistakes in its output, and use the modified output as input for the next round of training.

## Acknowledgements

This work has been made possible in part by CZI grant DAF2021-239849 and grant DOI (10.37921/244803gjbgup) from the Chan Zuckerberg Initiative DAF, an advised fund of Silicon Valley

Community Foundation. The work was also supported by NIH grant R35GM139622 and NSF grant MCB-2016090. We thank Mara Casebeer, Thias Boesen, and the members of the Jun Lab for testing and debugging napari-MM3. We also thank Kevin Cutler for helping us to install Omnipose and run it on our data.

## Additional information

### Funding

| Funder | Grant reference number | Author |
|---|---|---|
| Chan Zuckerberg Initiative | NPA-0000000033 | Suckjoon Jun |
| National Institute of General Medical Sciences | R35GM139622 | Suckjoon Jun |
| National Science Foundation | MCB-2016090 | Suckjoon Jun |
| Chan Zuckerberg Initiative | DAF2021-239849 | Suckjoon Jun |
| Chan Zuckerberg Initiative | 10.37921/244803gjbgup | Suckjoon Jun |

The funders had no role in study design, data collection, and interpretation, or the decision to submit the work for publication.

### Author contributions

Ryan Thiermann, Michael Sandler, Conceptualization, Data curation, Software, Formal analysis, Investigation, Visualization, Writing - original draft, Writing - review and editing; Gursharan Ahir, Guillaume Le Treut, Fangwei Si, Dongyang Li, Data curation, Software, Writing - review and editing; John T Sauls, Jeremy Schroeder, Conceptualization, Software, Visualization, Methodology, Writing - original draft, Writing - review and editing; Steven Brown, Conceptualization, Software, Visualization, Methodology; Jue D Wang, Supervision, Funding acquisition, Writing - review and editing; Suckjoon Jun, Conceptualization, Supervision, Funding acquisition, Visualization, Writing - original draft, Writing - review and editing

### Author ORCIDs

Ryan Thiermann ⓘ http://orcid.org/0000-0002-0348-4181
Jeremy Schroeder ⓘ http://orcid.org/0000-0001-8829-8494
Jue D Wang ⓘ http://orcid.org/0000-0003-1503-170X
Suckjoon Jun ⓘ http://orcid.org/0000-0002-0139-4297

Reviewer #1 (Public Review): https://doi.org/10.7554/eLife.88463.4.sa1
Reviewer #2 (Public Review): https://doi.org/10.7554/eLife.88463.4.sa2
Author response https://doi.org/10.7554/eLife.88463.4.sa3

## Additional files

### Supplementary files
• MDAR checklist

### Data availability

No new raw data was generated for this manuscript. Raw imaging data (previously analyzed in *Si et al., 2019*) has been stored in Dryad. Processed data is available on GitHub, (*Thiermann, 2024a*, copy archived at *Thiermann, 2024b*). Source code used to generate Figure 4 is also available on GitHub (*Thiermann et al., 2024a*, copy archived at *Thiermann et al., 2024b*).

The following previously published dataset was used:

| Author(s) | Year | Dataset title | Dataset URL | Database and Identifier |
|---|---|---|---|---|
| Thiermann R, Sandler M, Ahir G, Sauls JT, Schroeder JW, Brown SD, Le Treut G, Si F, Li D, Wang JD, Jun S | 2024 | Data for: Tools and methods for high-throughput single-cell imaging with the mother machine | https://doi.org/10.5061/dryad.2fqz612xd | Dryad Digital Repository, 10.5061/dryad.2fqz612xd |

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
