## [Editor Report · eLife assessment]

This article provides a review and test of image-analysis methods for bacteria growing in the 'mother-machine' microfluidic device, introducing also a new graphical user interface for the computational analysis of mother-machine movies based on the 'Napari' environment. The tool allows users to segment cells based on two previously published methods (classical image transformation and thresholding as well as UNet-based analysis), with **solid** evidence for their robust performance based on comparison with other methods and use of datasets from other labs. While it was difficult to assess the user-friendliness of the new GUI, it appears to be **valuable** and promising for the field.

---

## [Referee Report · Reviewer #1 (Public Review)]

The authors aim to develop an easy-to-use image analysis tool for the mother machine that is used for single-cell time-lapse imaging. Compared with related software, they tried to make this software more user-friendly for non-experts with a design of "What You Put Is What You Get". This software is implemented as a plugin of Napari, which is an emerging microscopy image analysis platform. The users can interactively adjust the parameters in the pipeline with good visualization and interaction interface.

Strengths:

- Updated platform with great 2D/3D visualization and annotation support.

- Integrated one-stop pipeline for mather machine image processing.

- Interactive user-friendly interface.

- The users can have a visualization of intermediate results and adjust the parameters.

Weaknesses:

- Based on the presentation of the manuscript, it is not clear that the goals are fully achieved.

- Although there is great potential, there is little evidence that this tool has been adopted by other labs.

- the diversity of datasets used in this study is limited.

- Some paragraphs in the Discussion section are like blogs with general recommendations. Although the suggestions look pretty useful, it is not the focus of this manuscript. It might be more appropriate to put it in the GitHub repo or a documentation page. The discussion should still focus on the software, such as features, software maintenance, software development roadmap, and community adoption.

A discussion of the likely impact of the work on the field, and the utility of the methods and data to the community.

- The impact of this work depends on the adoption of the software MM3. Napari is a promising platform with an expanding community. With good software user experience and long-term support, there is a good chance that this tool could be widely adopted in the mother machine image analysis community.

- The data analysis in this manuscript is used as a demo of MM3 features, rather than scientific research.

---

## [Referee Report · Reviewer #2 (Public Review)]

The authors present an image-analysis pipeline for mother-machine data, i.e., for time-lapses of single bacterial cells growing for many generations in one-dimensional microfluidic channels. The pipeline is available as a plugin of the python-based image-analysis platform Napari. The tool comes with two different previously published methods to segment cells (classical image transformation and thresholding as well as UNet-based analysis), which compare qualitatively and quantitatively well with the results of widely accessible tools developed by others (BACNET, DelTA, Omnipose). The tool comes with a graphical user interface and example scripts, which should make it valuable for other mother-machine users, even if this has not been demonstrated yet.

The authors also add a practical overview of how to prepare and conduct mother-machine experiments, citing their previous work, referring to detailed instructions on their github page, and giving more advice on how to load cells using centrifugation.

Finally, the authors emphasize that machine-learning methods for image segmentation reproduce average quantities of training datasets, such as the length at birth or division. Therefore, differences in training can propagate to differences in measured average quantities. This result is not surprising but good to remember before interpreting absolute measurements of cell shape.

---

## [Author Response]

The following is the authors’ response to the previous reviews.

After revision, I only have a few remaining remarks:l. 180 The authors write: We were able to process all 4 datasets with minimal adjustments to the default parameter values (Methods).But they still don't indicate how they vary parameters and how important this is for success or how this affects absolute measurements such as average cell length. Could they give a table of parameter values and some sense of sensitivity for any future user?

We thank the reviewer for the suggestion. We see how this info is valuable for the user. We’ve added a table with the parameter values used for processing each dataset in the supplemental information, along with the default parameters for reference (lines 476 - 496). In that section we also discuss which parameters may affect the output measurements of cell size, etc.

l. 192-193 They write 'The software performed well on BACMMAN, molyso and MoMa datasets.' Naming the datasets after the analysis methods used in the original papers could be confusing, as they analyse data with MM3. Not sure how best to resolve this, maybe using first author names instead.

We thank the reviewer for pointing this out. We now refer to them with the first author names.

Related to the request of ref. #1 for a video tutorial, the video currently displayed under the github readme.md section 'Usage guide' is not functional. And the video at the top of the same page is very short with minimal information.

We thank the reviewer for letting us know the tutorial video was not functional. We’ve tested it on Linux, Mac and Windows machines on both Firefox and Chrome. We were not able to reproduce any problems for the video - could they let us know what browser / OS was used and any other specifics? If it’s easier, we can be reached through the Github page as well.